# A double-blind, placebo-controlled, randomized, multi-centre, phase III study of MLC901 (NeuroAiD™II) for the treatment of cognitive impairment after mild traumatic brain injury

Pavel I. Pilipenko[1†], Anna A. Ivanova[2], Yulia V. Kotsiubinskaya[3], Vera N. Grigoryeva[4], Alexey Y. Khrulev[4], Anatoly V. Skorokhodov[5], Maxim M. Gavrik[6], Nona N. Mkrtchan[7], Marek Majdan[8], Peter Valkovic[9], Daria Rabarova[8], Suzanne Barker-Collo[10], Kelly Jones[11], Valery L. Feigin[11]*

1 Novosibirsk State Medical University, Novosibirsk, Russia, 2 Municipal Polyclinic № 106 of St. Petersburg, St. Petersburg, Russia, 3 X7 Research, St. Petersburg, Russia, 4 Nizhny Novgorod Regional Clinical Hospital named after N.A. Semashko, Nizhny Novgorod, Russia, 5 Ivanovo Regional Clinical Hospital, Ivanovo, Russia, 6 Scientific Research Center Eco Safety LLC, St. Petersburg, Russia, 7 Medical Technologies Ltd, St. Petersburg, Russia, 8 Institute for Global Health and Epidemiology, Trnava University, Trnava, Slovakia, 9 2nd Department of Neurology, Faculty of Medicine, Comenius University in Bratislava, Bratislava, Slovakia, 10 School of Psychology, The University of Auckland, Auckland, New Zealand, 11 National Institute for Stroke and Applied Neurosciences, School of Clinical Sciences, Faculty of Health and Environmental Sciences, Auckland University of Technology, Auckland, New Zealand

† deceased
* valery.feigin@aut.ac.nz

## Abstract

### Introduction

About half of the world population will suffer from a traumatic brain injury (TBI) during their lifetime, of which about 90% of cases are mild TBI. Although up to 40% of adults with mild TBI experience persistent functional deficits, there is no proven-effective treatment to facilitate recovery after it.

### Methods and analysis

This randomized placebo-controlled multi-centre study was aimed to examine the efficacy of herbal supplement MLC901 on complex attention following mild TBI at 6 months post-randomisation, as a primary outcome measured by CNS Vital signs (CNS-VS). Adults aged 18–65 years, who were 1–12-months post-mild TBI and experienced cognitive impairment, were randomly assigned to receive either MLC901 two capsules (0.4g/capsule) or placebo three times a day for 6 months using centralized stratified permuted block randomization. Secondary outcomes: Rivermead Post-Concussion Symptoms Questionnaire (RPQ; neurobehavioral sequelae); Health Related Quality of Life (QOLIBRI); Hospital Anxiety and Depression Scale (HADS); and safety. Mixed effects models of repeated measures with intention to treat

**Data availability statement:** The study sponsor (Moleac Pte Ltd) is committed to provide the study datasets upon written request to Dr Sylvain Durrleman at sylvain.durrleman@mole-ac.com. Should the reasonable request for data sharing be raised, data access to all data used by the authors of this article will be provided without restriction.

**Funding:** This study was financially supported by Moleac Pte Ltd (https://moleac.com) in the form of an unrestricted educational grant. No additional external funding was received for this study. The funder had no role in study design, data collection and analysis, decision to publish, or preparation of the manuscript.

**Competing interests:** The authors would like to declare the following patents associated with this research: The MLC901 supplement that has been tested in the trial has a trade name NeuroAidTMII. This supplement has been granted two patents: EP2838545B1: This patent describes the use of Neuroaid2 components (including the nine herbal ingredients) for activating sarcolemmal KATP channels. US10463707B2: This patent details a process for preparing a mixture of herbal extracts from NeuroAid II components. This does not alter our adherence to PLOS ONE policies on sharing data and materials.

analysis were employed. A Least Square Mean Difference (LSMD) from baseline to 3-, 6-, and 9-month follow-up was calculated with 95% confidence intervals (CI).

## Results

In the analysis, 182 participants (47.8% females) were included. Multivariable mixed effects model analysis did not reveal significant improvements in complex attention (LSMD = −1.18 [95% CI −5.40; 3.03; p = 0.58]) and other cognitive domains at 6 months in the MLC901 group compared to the Placebo group. There were significant improvements in RPQ, QOLIBRI, anxiety and depression in the MLC901 group compared to the Placebo group at 6 and 9 months (LSMD −4.36 [−6.46; −2.26] and −4.07 [−6.22; −1.92], 4.84 [1.58; 8.10] and 3.74 [0.44; 7.03], −1.50 [−2.29; −0.71 and −0.96 [−1.84; −0.08], −1.14 [−1.92; −0.35] and −1.14 [−1.94; −0.34]), respectively. MLC901 tested was proven safe.

## Conclusions

Although the 6-month treatment with MLC901 did not result in a statistically significant difference with placebo for CNS-VS measurement of cognitive domains in individuals with mild TBI, the study showed a clinically and statistically significant improvement in all clinical scales assessed by the investigators.

## Trial registration

ClinicalTrials.gov identifier NCT04861688.

## Introduction

One of the main causes of disability and mortality among young adults worldwide is traumatic brain injury (TBI) [1,2]. An estimated 50% of people have had a TBI at some point in their life. Every year, the incidence of new TBIs is about 50 million, and this number is rising worldwide [2,3]. The annual global cost of TBI is already immense and estimated to be around 400 billion US dollars [2].

The two main causes of brain injury after head trauma are (i) diffuse axonal injury resulting from rotational forces as the brain rotates within the skull, and (ii) mechanical impact of the brain on bony surfaces of the skull or on objects penetrating the skull. As a result of injury, brain cells can be damaged or die, affecting the functioning of areas that they help to control (e.g., causing neurological/cognitive deficits). The temporal and basal-frontal regions of the brain, which are both linked to cognitive function, are the most commonly affected areas in closed TBI [4,5].

Around 90% of all TBI cases are mild TBI, and persistent cognitive deficits have been reported to occur in up to half of adults following a mild TBI and can profoundly impact a person's day-to-day functioning, often affecting their ability to return to work, or impacting their capacity to engage in independent living [5,6]. Cognitive flexibility, executive functioning, and complex attention deficiencies are the most prevalent

types of cognitive impairments. Many people recover spontaneously from brain injuries, especially mild ones, because the brain promotes healing of the affected area by mobilizing parts of central nervous system (CNS) that are still functional. However, often people continue to suffer from cognitive, emotional and physical deficits due to the insufficiency of this natural spontaneous process of brain self-repair [7–9].

Despite the huge and increasing burden from mild TBI, there is still no proven-effective pharmacological treatment to improve post-TBI cognitive functioning and further research into potential new interventions is needed. There is increasing evidence that the self-repair mechanisms involved after brain injury are multiple and complex involving various biological pathways and cellular changes. Also, it is unlikely that a single molecule that selectively acts on a specific target could be effective. Thus, combination therapies such as Chinese multi-herbal medicines including multiple active substances may provide a valuable resource in such a search for safe and effective therapy [10].

NeuroAiD™II (MLC901) is a botanical product, derived from traditional Chinese medicine, and containing extracts from 9 herbal ingredients. It is a reduced ingredient version of a predecessor product, MLC601 developed from a traditional Chinese medicine, Danqi Piantang Jiaonang, for stroke recovery, and has long been used in many people, notably to facilitate functional recovery of patients in the post-acute phase of stroke. The herbal components of MLC601 and MLC901 are the same. In in-vitro and animal experiments, both products have been shown to protect brain cells from dying after injury, and to stimulate generation of new neural cells, connections and pathways [11–14]. In a recent pilot placebo-controlled randomized trial conducted in New Zealand, MLC901 demonstrated significant improvement in complex attention and executive functioning in individuals who had experienced some cognitive impairment after mild TBI [15]. That pilot trial informed the design of the current study.

## Methods

### Overview of design

The **S**afety and effic**A**cy of **M**LC901 in cognitive recovery post tra**U**matic B**RA**in **I**njury (SAMURAI) study was a phase III double-blind, placebo-controlled, randomized multicentre clinical trial in five cities/centres in Russia. The study protocol and methodology has been previously published [16]. The primary aim of the study was to determine the cognitive benefit of using NeuroAiD™II (MLC901) for a treatment of 6 months, compared to placebo, in adult individuals who had recently suffered a mild-TBI, and to assess its safety.

The primary efficacy outcome of the study was the measure of complex attention at month 6, one of the numerous parameters included in the battery of measures assessed by using the CNS-Vital signs (CNS-VS) tool, an online auto-administered computerized cognitive test [17,18]. Secondary outcomes included other parameters of the CNS-VS (executive functioning, processing speed, memory [visual and verbal] and reaction time), as well as four commonly used investigator assessment clinical outcome scales for TBI patients, namely post-concussion symptoms (as measured by Rivermead Post-Concussion Symptoms Questionnaire [RPQ]) [19], Health Related Quality of Life After Brain Injury (QOLIBRI) [20], anxiety and depression (as measured by the Hospital Anxiety and Depression Scale [HADS]) [21,22], and adverse events.

The investigational product, active or placebo capsules, were identical in colour, shape and taste, and the investigators and study team remained blind to group allocation throughout the trial. An independent Data Safety and Monitoring Committee (DSMC) ensured study oversight and protection of patients' safety, by reviewing overall data during its meetings, while remaining blinded to treatment allocation. No interim analysis was performed.

The study was conducted in accordance with the ethical principles of good clinical practice, the Declaration of Helsinki [23] and all local regulations. SAMURAI was approved by the Ethics Committee of the Ministry of Health of the Russian Federation on 9 March 2021 (Dossier Ref#58074, meeting's protocol #268) and all local Ethics Committees. SAMURAI is registered on ClinicalTrials.gov identifier NCT04861688. Only study participants who provided a written informed consent dated and signed at the presence of the study investigator were randomised. The first study participant was randomized on 18/08/2021, and the last one – on 29/04/2022.

## Power calculations

The required sample size for the trial was based on the results of the pilot double-blind, place-controlled randomized clinical trial in New Zealand (n = 78) evaluating the effect of MLC901 versus placebo using CNS-VS [15]. It was estimated that a sample of 182 participants (with 1:1 group ratio) would provide 80% statistical power (two sided α = 0.05, β = 0.20) to detect a clinically significant 10 points difference [24] (SD = 20) between placebo and MLC901 in the change from baseline to month 6 for the CNS-VS complex attention score, assuming 30% non-compliance/lost to follow-up (for details see Table S1 in S2 File of supplementary materials).

## Study participants and treatments

The SAMURAI trial included 182 adults of 18–65 years of age who experienced a mild TBI in the past 1–12 months, had cognitive functioning difficulties as indicated by a score of >30 on the Cognitive Failures Questionnaire [25] (consistent with criteria in the pilot study) [15], and gave informed consent to participate in the study. Mild TBI was defined according to the WHO criteria (Glasgow Coma Score 13–15 as assessed on scene, on admission and over next 3 days; loss of consciousness for up to 30 minutes; being dazed and confused at the time of injury or post-TBI amnesia of < 24 hours duration). Exclusion criteria: (1) co-existing severe co-morbidity, including end-stage renal failure, spinal cord injury, significant substance abuse, severe liver disease, significant mental illnesses, diabetes requiring insulin injections, severe agitation, advanced cancer or other severe conditions with life expectancy of less than 5 years, as judged by the study investigator (neurologist); (2) current participation in another clinical trial within 30 days; (3) women who were pregnant or who had a positive urine pregnancy test or breast-feeding; and (4) not fluent in Russian language or have aphasia/dysphasia.

Eligible individuals were randomized to receive either the NeuroAiD™II (MLC901) two capsules (0.4g/capsule) or matched placebo orally three times a day for 6 months using 1:1 stratified permuted block randomization (stratified by study centre, time since injury [1–3 months/4–12 months] and gender).

## Outcomes

The primary outcome measure was complex attention, as measured by an online CNS Vital signs computerized cognitive test [17,18], at 6 months post randomization. CNS-VS is a computerized neurocognitive test battery that was developed as a routine clinical screening instrument [17]. During CNS-VS testing (for measuring primary and secondary cognitive outcomes), after the patient completes the seven cognitive tests, the system automatically calculates scores for domains. Speed and accuracy on six tests were used to calculate the level of functioning across the following cognitive domains: complex attention, executive function, verbal memory, visual memory, processing speed, and reaction time. Raw scores were transformed to standard scores, with a mean of 100 and standard deviation (SD) of 10, based on normative data to account for age and gender effects using an integrated algorithm. Scores <90 indicate below average levels of functioning, with higher scores indicative of better cognitive functioning. The psychometric characteristics of the tests in the CNS-VS battery are very similar to the characteristics of the conventional neuropsychological tests upon which they are based [26,27]. CNS-VS has demonstrated good discriminant and concurrent validity with conventional neuropsychological tests [17] and is sensitive to impairments across TBI severity, with evidence of good test–retest reliability [18,24].

The RPQ [19] assesses neurobehavioral sequelae and consists of two subscales including the RPQ3, which includes symptoms of headaches, dizziness and nausea, and the RPQ13 comprising 13 other common symptoms such as restlessness, noise and light sensitivity, sleep disturbance, blurred vision and balance difficulties. Participants are to state the extent to which they experience each symptom in comparison to the time before accident, on a 5-point scale ranging from 0 (not experienced) to 4 (severe problem). The two subscales have revealed good test-re-test reliability and adequate external construct validity [28,29]. A total score ranges from 0 to 64 with higher values indicating greater symptom severity.

The QOLIBRI [20] is an internationally validated tool to assess quality of life after brain injury [30]. It contains two parts. The first part assesses satisfaction with health-related quality of life and is composed of six overall items and 29 items

allocated to four subscales: thinking, feelings, autonomy, and social aspects. The second part, devoted to "bothered" questions, is composed of 12 items in two subscales: negative feelings and restrictions. QOLIBRI total scores <60 indicate low or impaired health-related quality of life [31]. The QOLIBRI showed good construct validity in the TBI group [32].

Mood was assessed by the HADS [22]. The scale has been widely used for assessing levels of anxiety and depression in patients with medical problems including TBI [33]. The scale consists of 14 statements (e.g., I feel tense or 'wound up') that the participant is asked to rate in regards how they have been feeling in the past week, yielding separate subscale scores for anxiety and depression. Scoring for each item ranges from zero to three, with three denoting highest anxiety or depression level. The subscale scores range from 0–21 (0–7 normal, 8–10 mild, 11–14 moderate and 15–21 severe). The measure has demonstrated good test-retest reliability [34] and good sensitivity and specificity [35].

Adverse events were monitored at each follow-up assessment visit. An adverse event was defined as any untoward medical occurrence in a study participant that does not necessarily have a causal relationship with the treatment. Assessments of primary and secondary outcomes were completed at baseline and 1-, 3-, 6-, and 9-month follow-up.

## Statistical analyses

Socio-demographic and clinical characteristics of the study participants at baseline were evaluated by tests of difference. For the analysis of primary and secondary endpoints, a mixed effects model (PROC MIXED) [36] was used with adjustments for baseline and potential covariates, with the participants and sites used as the random effects. Model selection was undertaken with each outcome using standard selection heuristics, with inclusion in the model variables clinically considered as important covariates. Covariates were selected based on improving the overall efficiency of the model. Regardless, baseline value of the outcome variable, age, gender, time since injury (1–3 months/4–12 months) and study center were included as covariates in the mixed effects model. Separate mixed models were constructed for each of the outcome parameters and their dependent variable was the respective outcome parameter change from baseline. The analysis used a repeated measure design with all post baseline timepoints used as repeated effect in each model, while treatment group and its interaction with timepoints along with parameter baseline value, participant age, gender, time since injury (dichotomized into two groups: 1–3 months/4–12 months) and study centers were included as fixed effects in each model. Estimates obtained from the model (Least Squares Means) for the parameters change from baseline at month 6 (primary timepoint) and month 9 were provided with their associated 95%CI and p-values to conclude about treatment efficacy.

Descriptive statistics were used to describe changes in complex attention and other outcome measures in the MLC901 and Placebo groups, as well as mean values and SDs for non-cognitive outcomes obtained by the method of least square mean difference (LSMD) from the mixed effects model. Mixed effects models of repeated measures with intention to treat analysis were employed, with the primary outcome time-point of 6 months. LSMD from baseline to 3-, 6-, and 9-month follow-up was calculated with 95% confidence intervals (CI). The Wilcoxon test was used to evaluate differences in ordinal items of outcomes. Safety and tolerability were assessed by the frequency and nature of any potential adverse events recorded. T-tests and Chi-2 tests were used for the quantitative and the categorical variables, respectively. Levels of adherence to the treatment regimen were determined based on the self-reported number of capsules not taken. Analyses were based on the intention to treat principle.

## Results

The trial results were subject of an abstract presented at the World Congress of Neurology in 2023 [37]. Of the 811 individuals assessed for eligibility, 182 participants (22%) were enrolled and randomized into the study (Figure S1 in S2 File of supplementary materials), with the retention rate of 98% at 9-month follow-up (one participant was lost to follow-up and three participants decided to withdraw from the study for unknown reasons). Table 1 indicates that there were no statistically significant differences between the two groups with respect to demographic and other baseline variables. The majority of mild TBIs were caused by falls (61%−62%). The median time from TBI to inclusion was 3 months.

**Table 1. Baseline demographic and medical characteristics of the study participants.**

| Characteristics | MLC901 Group N = 92 | Placebo Group N = 90 |
|---|---|---|
| Age (yrs) mean (SD) | 40.6 (14.2) | 40.1 (12.0) |
| Males (%) | 50.0 | 47.8 |
| Caucasian ethnicity (%) | 100.0 | 97.8 |
| Tertiary education or above (%) | 50.0 | 62.2 |
| Months since injury (median) | 3.0 | 2.9 |
| Full-time pre-TBI work (%) | 87.0 | 94.5 |
| Married/Partner (%) | 69.6 | 80.0 |
| Mechanism of TBI injury (%) Fall Motor vehicle accident Violence Sport trauma Other | 60.9 14.1 9.8 12.0 3.3 | 62.2 13.3 11.1 6.7 6.7 |
| Prior TBI (%) | 32.6 | 40.0 |
| Other previous injuries sustained (%) | 47.8 | 55.5 |
| Baseline CNS-Vital score below average (%) | 57.6 | 52.2 |

Previous history of TBI was reported in 32.6% of the MCL901 group and 40.0% of the Placebo group participants. About half of the randomized participants in both groups were females.

Twenty-seven participants in the MLC901 group (29.3%) and 34 participants in the Placebo group (37.8%) reported side effects during the 6 months post-randomization (Table 2). In regard to the most common side effect, in the intervention group, eight participants reported gastrointestinal symptoms (8.8%), as compared to three participants in the control group (3.3%). No serious adverse events (death, hospitalization, disability) were reported in either group. As measured by the returned caps counting, overall adherence to study treatment varied between 99.1% (SD 2.5) for MLC901 and 98.4% (SD 10.7) for placebo at 1 month, 99.7% (SD 3.8) for MLC901 and 98.6% (SD 4.5) at 3-months and 98.9% (SD 3.9) for MLC901 and 99.5% (SD 4.3) for placebo at 6 months.

The assessment of cognitive impairment made by the patient by using the CNS-VS tool did not reveal a statistically significant difference between placebo and MLC901 (Table 3; Fig 1; Table S3 in S2 File of supplementary materials). This was the case for the primary outcome (change from month 6 to baseline in complex attention (LSMD = −1.18 [95% CI −5.40, 3.03], p = 0.58), as well as for the other CNS-VS derived cognitive parameters (executive functioning, visual memory, verbal memory, processing speed, or reaction time).

Participants randomized to receive MLC901 had statistically and clinically significant improvements in post-concussion symptoms (Table S4 in S2 File of supplementary materials), quality of life, anxiety and depression at 6- and 9-months post-randomization compared to participants randomized to receive placebo (Table 3; Figs 2–5). Importantly, the improvement continued up to 9 months of follow-up although active treatment was stopped at 6 months after randomization. By the 9-month follow-up the RPQ score improved in the MCL901 group by 47% (95% CI 41%, 53%), while in the Placebo group it improved by only 29% (95% CI 22%, 35%), QOLIBRI score improved in the MLC901 group by 22% (95% CI 18%, 26%) and only 14% (96% CI 11%, 18%) in the Placebo group, HADS anxiety score improved in MLC901 group by 49% (95% CI 41%, 57%) and only 42% (95% CI 33%, 51%) in the Placebo group, and depression score reduced by 48% (95% CI 39%, 57%) in the MLC901 group and only 31% (95% CI 22%, 40%) in the Placebo group. Pre-determined subgroup sensitivity analysis with exclusion of missing values, outliers by age, sex at birth, history of cancer, previous TBI, and time since mild TBI event onset did not reveal any significant difference in cognitive outcomes between the groups (these results are not shown).

**Table 2. Adverse effects in the MLC901 and Placebo groups (n, %) at 9 months post-randomisation.**

| Adverse events | MLC901 Group N=92 | Placebo Group N=90 |
|---|---|---|
| Headache | 8 (8.9%) | 2 (2.2%) |
| Fatigue | 6 (6.7%) | 3 (3.3%) |
| Respiratory tract infection viral | 5 (5.6%) | 4 (4.3%) |
| COVID-19 | 4 (4.4%) | 2 (2.2%) |
| Dizziness | 3 (3.3%) | 0 |
| Nausea | 1 (1.1%) | 2 (2.2%) |
| Abdominal discomfort | 0 | 2 (2.2%) |
| Abdominal pain | 0 | 2 (2.2%) |
| Alanine aminotransferase increased | 1 (1.1%) | 1 (1.1%) |
| Aspartate aminotransferase increased | 1 (1.1%) | 1 (1.1%) |
| Bronchitis | 1 (1.1%) | 1 (1.1%) |
| Diarrhoea | 0 | 2 (2.2%) |
| Nasopharyngitis | 1 (1.1%) | 1 (1.1%) |
| Respiratory tract infection | 1 (1.1%) | 1 (1.1%) |
| Abdominal distension | 1 (1.1%) | 0 |
| Anterograde amnesia | 0 | 1 (1.1%) |
| Asthenia | 0 | 1 (1.1%) |
| Biliary colic | 0 | 1 (1.1%) |
| Cholelithiasis | 0 | 1 (1.1%) |
| Decreased appetite | 0 | 1 (1.1%) |
| Dermatitis acneiform | 1 (1.1%) | 0 |
| Disturbance in attention | 1 (1.1%) | 0 |
| Eye injury | 1 (1.1%) | 0 |
| Haematuria | 0 | 1 (1.1%) |
| Numbness in arms or legs | 0 | 1 (1.1%) |
| Hyposmia | 0 | 1 (1.1%) |
| Hypothyroidism | 0 | 1 (1.1%) |
| Increased appetite | 0 | 1 (1.1%) |
| Insomnia | 0 | 1 (1.1%) |
| Irritable bowel syndrome | 1 (1.1%) | 0 |
| Laryngitis | 1 (1.1%) | 0 |
| Memory impairment | 1 (1.1%) | 0 |
| Osteoarthritis | 0 | 1 (1.1%) |
| Otitis media acute | 1 (1.1%) | 0 |
| Photophobia | 1 (1.1%) | 0 |
| Rhinitis | 1 (1.1%) | 0 |
| Rhinitis allergic | 0 | 1 (1.1%) |
| Sleep disorder | 0 | 1 (1.1%) |
| Spinal pain | 0 | 1 (1.1%) |
| Taste disorder | 0 | 1 (1.1%) |
| Tinnitus | 1 (1.1%) | 0 |
| Tonsillitis | 1 (1.1%) | 0 |
| Tremor | 1 (1.1%) | 0 |
| Vertigo | 0 | 1 (1.1%) |
| Viral infection | 0 | 1 (1.1%) |
| Vision blurred | 1 (1.1%) | 0 |

**Table 3. Least square mean difference, with 95% CI, in primary and secondary outcomes between MLC901 group and Placebo group at 6 and 9 months after randomisation.**

| Outcomes | Least Square mean difference (95% CI) MLC901 vs Placebo | | | |
|---|---|---|---|---|
| | 6 months | *P* | 9 months | *P* |
| Cognitive functioning | −1.18 (−5.40 to 3.03) | *0.58* | −0.53 (−4.54 to 3.47) | *0.79* |
| Complex attention | −0.48 (−4.63 to 3.67) | *0.82* | 0.34 (−3.67 to 4.34) | *0.87* |
| Executive functioning | −2.81 (−7.67 to 2.04) | *0.25* | −0.96 (−5.89 to 3.98) | *0.70* |
| Visual memory | 0.64 (−3.27 to 4.56) | *0.74* | −1.60 (−6.58 to 3.39) | *0.53* |
| Verbal memory | 2.88 (−2.38 to 8.13) | *0.28* | 1.00 (−3.79 to 5.79) | *0.68* |
| Processing speed | −0.38 (−4.86 to 4.11) | *0.87* | 0.23 (−4.49 to 4.95) | *0.92* |
| Reaction time | | | | |
| Rivermead Post-Concussion Symptoms Questionnaire | −4.36 (−6.46 to −2.26) | *0.0001* | −4.07 (−6.22 to −1.92) | *0.0003* |
| Change in Health-related quality of life questionnaire | 4.84 (1.58 to 8.10) | *0.0038* | 3.74 (0.44 to 7.03) | *0.0264* |
| Hospital Anxiety and Depression Scale | −1.50 (−2.29 to −0.71) | *0.0003* | −0.96 (−1.84 to −0.08) | *0.0333* |
| Anxiety | −1.14 (−1.92 to −0.35) | *0.0050* | −1.14 (−1.94 to −0.34) | *0.0054* |
| Depression | | | | |

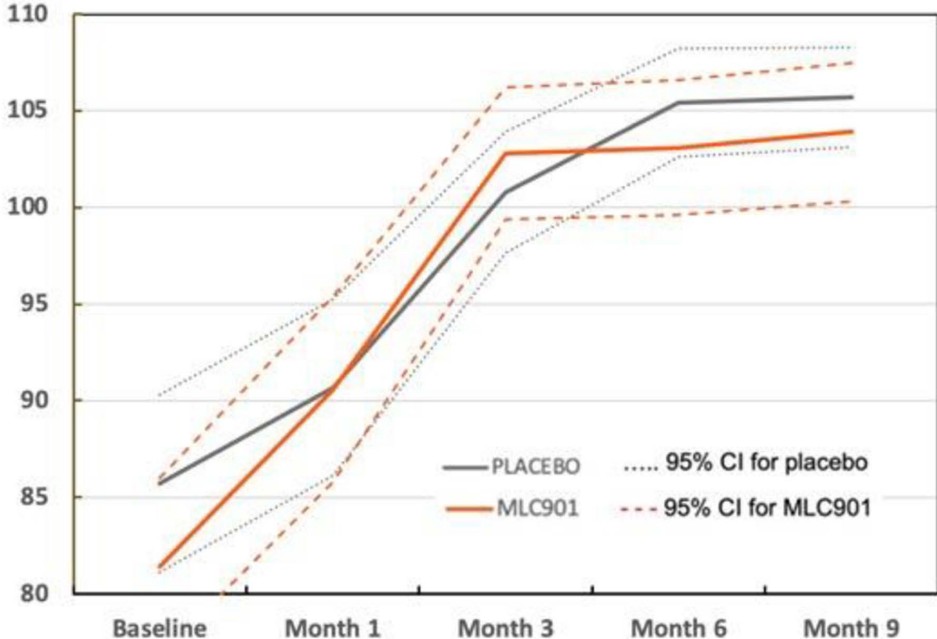

Least square mean difference at 6 months = -1.18 (95% CI -5.40 to 3.03); p=0.58. Overlapping confidence intervals (shown in dotted lines) indicate no statistically significant difference between MLC901 and Placebo groups across all follow-up time points.

**Fig 1. Changes in complex attention by CNS Vital Signs across the follow-up time points in MLC901 group compared to Placebo group across 9 months of follow-up period.**

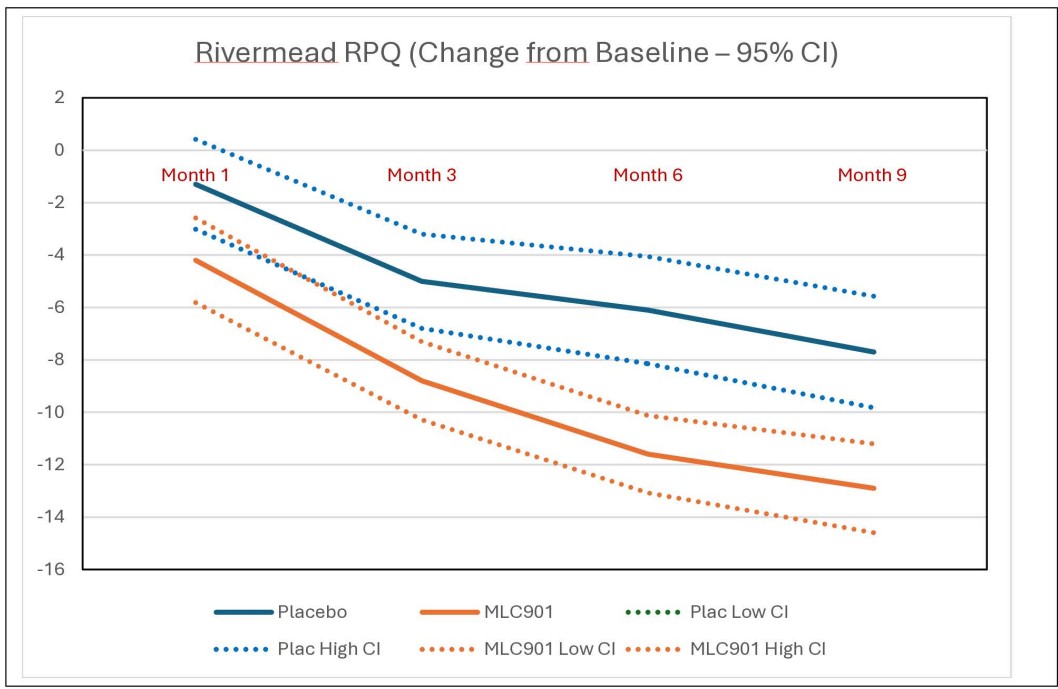

Least square mean difference at 6 months = 4.36 (-6.46 to -2.26); p = 0.0001. Dotted lines denote 95% confidence intervals.

**Fig 2. Changes in the Rivermead Scale across the follow-up time points in MLC901 group compared to Placebo group across 9 months of follow-up period.**

## Discussion

To the best of our knowledge this placebo-controlled randomized full-scale trial is the first to determine efficacy of MLC901 (2 capsules 3 times a day for 6 months) on cognitive functioning and post-mild TBI symptoms compared to a matched placebo in adults 16–65 years old with mild TBI. The trial failed to demonstrate a statistically significant difference in the CNS-VS computerized assessment of complex attention (Fig 6) and other CNS-VS measures between the MLC901 group and the Placebo group (Table S1 in S2 File of supplementary materials). However, the assessment of post-TBI symptoms using standard clinical non-cognitive outcome scales used in this setting has shown statistically significant difference at both 6 and 9 months in favour of MLC901. These findings were consistent for all clinical outcome scales used in the trial.

Analysis of common post-TBI somatic symptoms in our trial measured by the RPQ (headaches, dizziness, noise sensitivity, sleep disturbance, fatigue, irritability, depression, poor memory, poor concentration, blurred vision, light sensitivity, double vision, restlessness) [19], quality of life (as measured by the QOLIBRI) [20], anxiety and depression (as measured by the HADS) [22] showed clinically and statistically significant improvement in the MLC901 group compared to the Placebo group: LSMD = −4.36 (−6.46 to −2.26); 4.84 (1.58 to 8.10), and −1.50 (−2.29 to −0.71) and −1.14 (−1.92 to −0.35), respectively (Figs 1–4). Importantly, the improvements tended to continue and were even enhanced at 9 months of follow-up while the active treatment was stopped at 6 months post-randomization, suggesting that the treatment somehow corrected the pathophysiological mechanisms responsible for these symptoms and triggered a self-recovering process. Although it remains to be confirmed how long this recovery process could continue beyond 9 months in both groups, the significant improvements in non-cognitive secondary outcomes up to 9 months post-randomisation are of considerable clinical significance. Indeed, up to

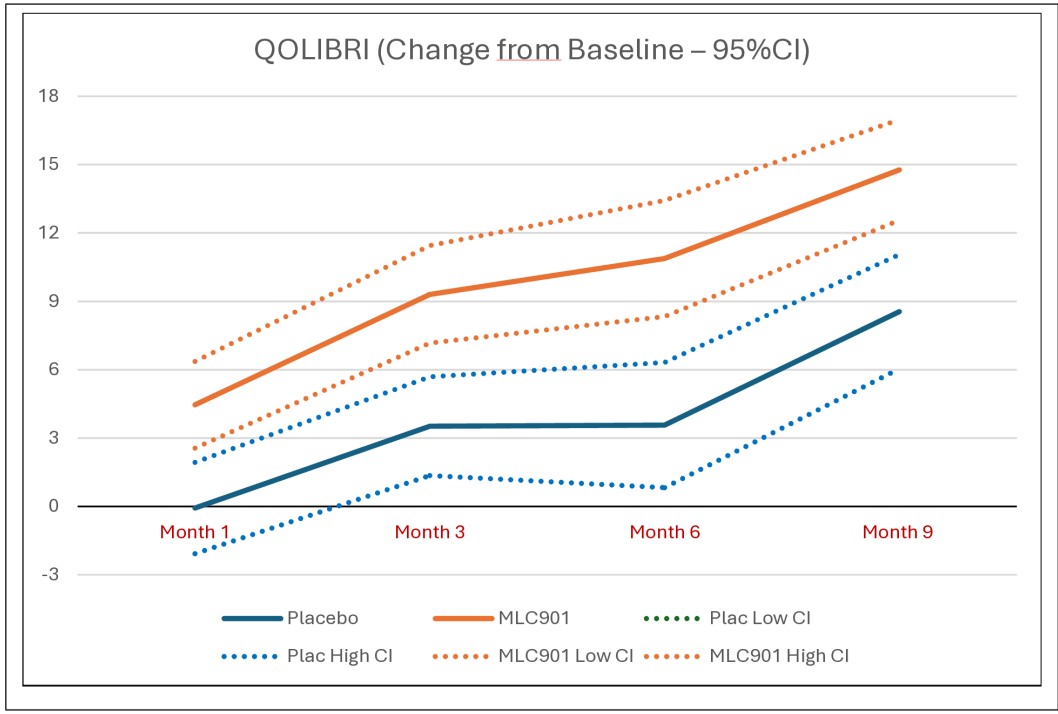

Least square mean difference at 6 months = 4.84 (1.58 to 8.10); p = 0.0038. Dotted lines denote 95% confidence intervals.

**Fig 3. Changes in the Quality of Life after Brain Injury measured across the follow-up time points in MLC901 group compared to Placebo group across 9 months of follow-up period.**

20% of people with mild TBI show persistent post-concussion symptoms [38], about one third of people with mild TBI experience reduced quality of life [30,39], and many of them suffer from anxiety or depression [40].

The trial also showed an excellent safety and tolerability profile of MLC901. The number of side effects (none of them being considered as serious adverse event [SAE]) was low, and they were more often observed in the Placebo group than in MLC901 group (Table 2). None of the side effects required discontinuation of the treatment. Moreover, adherence to the trial medications across the 6 months of the treatment was close to 100%.

Comparison of results of this trial with the previous pilot placebo-controlled trial of MLC901 in New Zealand [15] showed that, contrary to the finding of the earlier study, the full-scale trial did not show a statistically significant improvement in complex attention, as measured by the computerized CNS-VS tool. However, the CNS-VS tool is relatively complex to use, and there is very little experience of its use in Russia. It is therefore possible that some questions and instructions used in performing the tests were not done properly in the study in Russia.

This assumption is supported by the observation of statistically significant improvement in some important items of the commonly used and well validated in Russia, RPQ [41], specifically forgetfulness/poor memory (P = 0.008 at 6 months and P = 0.006 at 9 months of follow-up) and concentration function (P = 0.002 at 6 months and P = 0.049 at 9 months of follow-up) in the MLC901 group compared to the Placebo group. A cognitive sub-item of the RPQ, the speed of processing information (question "taking longer to think"), also demonstrated a statistically significant improvement) in the MLC901 group compared to the Placebo group (P = 0.009; for details see Table S2 in S2 File of supplementary materials). This is a

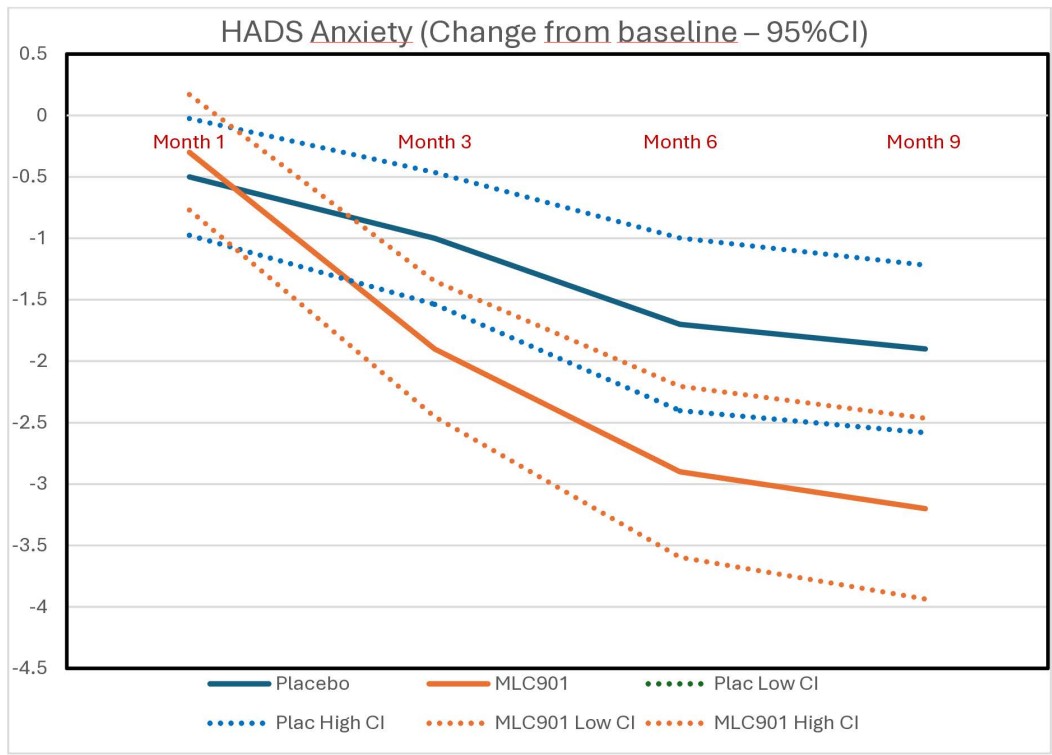

Least square mean difference at 6 months = -1.50 (-2.29 to -0.71); p = 0.0003. Dotted lines denote
95% confidence intervals.

**Fig 4. Changes in the anxiety score, as measured by the Hospital Anxiety and Depression Scale, across the follow-up time points in MLC901 group compared to Placebo group across 9 months of follow-up period.**

particularly important clinical outcome, because speed processing is shown to be one of the most important indicators of diffuse axonal injury and TBI severity [42,43]. The lack of statistical differences obtained for the CNS-VS endpoints in mild TBI subjects may also be related to the insufficient ability of the CNS-VS to detect subtle changes in cognition 6–9 months after a mild TBI, as there is evidence that long after a mild TBI, high-functioning young adults invoke a strategy of delaying their identification of targets in order to maintain, and facilitate, accuracy on cognitively demanding tasks [44].

Sensitivity analysis in various age and sex groups, different time points after TBI onset and randomization, and exclusion of outliers did not significantly change the positive treatment effect of MLC901 on post-concussion symptoms, quality of life, and mood, thus supporting the robustness of the results. These observations deserve further investigation.

Although our trial met most of the criteria for 'gold standard' trials (placebo-controlled, double-blinded, proper quality randomization technique, high generalizability of the trial results [wide inclusion and narrow exclusion criteria, heterogeneous study population with multi-centre-settings], fully-statistically powered with a relatively large sample size, and very low [<10%] attrition rate), it was not free from limitations. The main limitation of the trial was lack of use and absence of validation of the computerized CNS-VS test in Russia. CNS-VS is also only a cognitive screening tool, not validated for treatment evaluation. CNS-VS data was also missing for a small number of participants in both groups (5 in MLC901 group and 5 in Placebo group) and outliers in regard to missing data in some of the recruitment sites (i.e., the majority of missing data 8 [80%] came from a single site). However, sensitivity analyses with various assumptions, including exclusion of outliers, did

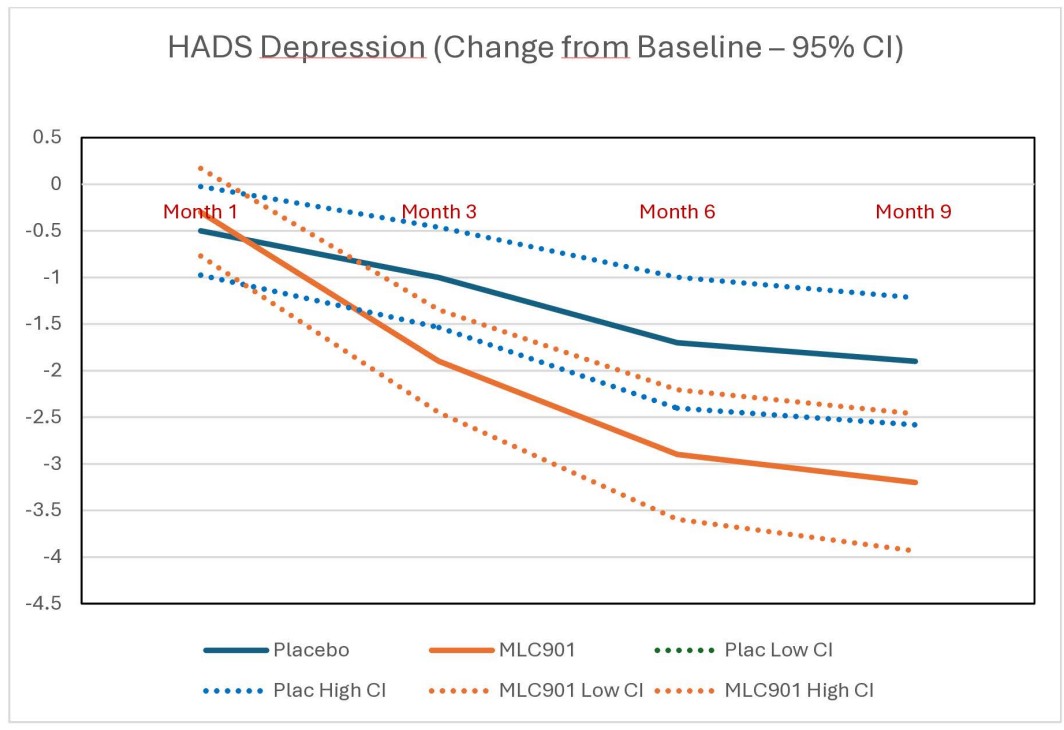

Least square mean difference at 6 months = -1.14 (-1.92 to -0.35); p= 0.0050. Dotted lines denote 95% confidence intervals.

**Fig 5. Changes in the depression score, as measured by the Hospital Anxiety and Depression Scale, across the follow-up time points in MLC901 group compared to Placebo group across 9 months of follow-up period.**

not significantly influence the neurocognitive test results. We also did not record adverse effects right after cessation of the experimental treatment (6 months post-randomization), but only at 9 months post-randomization. Moreover, due to COVID-19 pandemic during the trial and associated restrictions on face-to-face meetings, training of the research staff on the use of CNS-Vital Signs was done via virtual meetings, we had a limited number of face-to-face meetings with the collaborators to discuss practical aspects of the CNS-Vital Signs test, and some study participants had limited digital literacy that might had led to suboptimal use of this computerised self-assessment tool. These methodological challenges and some limitations of the computerized assessment of CNS-VS of Complex Attention and other CNS-VS cognitive parameters in Russia (including cultural and linguistic factors) may have influenced the negative results of these outcome measures in the trial. Finally, the study population in this trial was limited to Russian participants, which may restrict its generalizability. Future studies could benefit from including a more diverse sample to broaden the applicability of the findings.

In summary, although the trial was not able to detect treatment difference for the primary outcome - - possibly because of insufficient validation of the CNS-VS computerized test in Russia – it was highly positive for all other important clinical post-TBI outcomes. Given the perfect safety and tolerability of MLC901 and absence of other proven effective medications for treatment post-mild TBI symptoms, mood and health-related quality of life, a trial treatment with MLC901 in selected mild TBI adults may be worth considering for reducing post-concussion symptoms, anxiety/depression and improving health-related quality of life. Adapting cognitive testing tools to different cultural contexts and exploring the long-term effects of MLC901 on non-cognitive outcomes are also warranted.

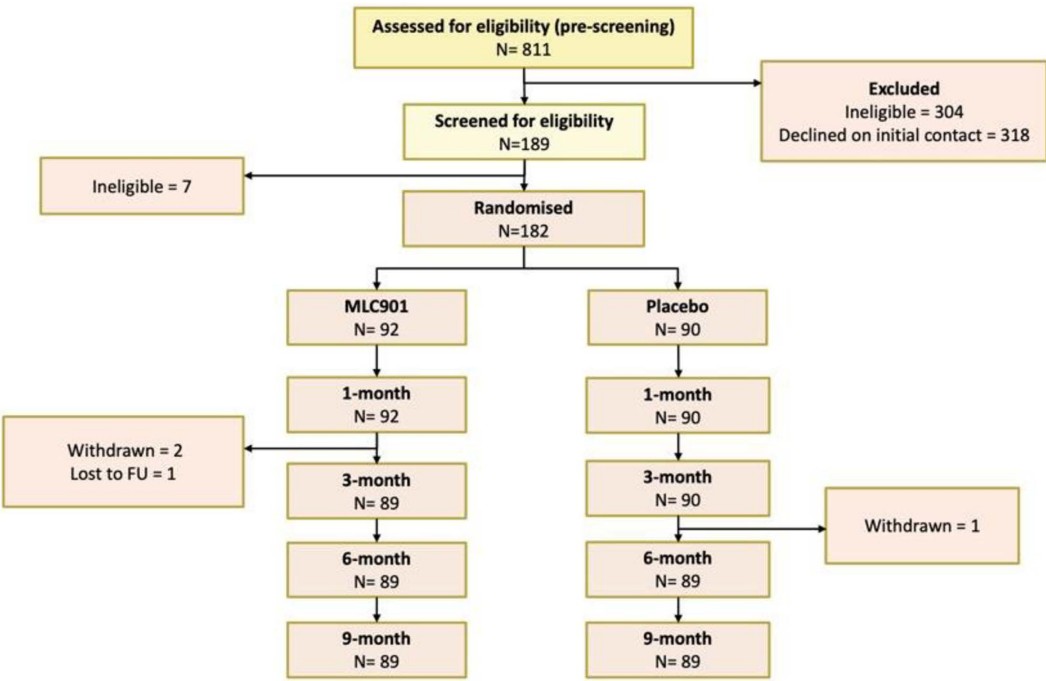

**Fig 6. CONSORT diagram of study participants.**

## Supporting information

**S1 File. Study protocol.**
(PDF)

**S2 File. Supplementary materials.**
(PDF)

**S3 File. CONSORT-2010-checklist.**
(DOC)

## Author contributions

**Conceptualization:** Valery L. Feigin.

**Writing – original draft:** Pavel I. Pilipenko, Valery L. Feigin.

**Writing – review & editing:** Anna A. Ivanova, Yulia V. Kotsiubinskaya, Vera N. Grigoryeva, Alexey Y. Khrulev, Anatoly V. Skorokhodov, Maxim M. Gavrik, Marek Majdan, Peter Valkovic, Daria Rabarova, Suzanne Barker-Collo, Kelly Jones.

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
