## [Decision Letter · Decision Letter 0]

PONE-D-24-36443A DOUBLE-BLIND, PLACEBO-CONTROLLED, RANDOMIZED, MULTI-CENTRE, PHASE III STUDY OF MLC901 (NEUROAID IITM) FOR THE TREATMENT OF COGNITIVE IMPAIRMENT AFTER MILD TRAUMATIC BRAIN INJURYPLOS ONE

Dear Dr. Feigin,

Thank you for submitting your manuscript to PLOS ONE. After careful consideration, we feel that it has merit but does not fully meet PLOS ONE’s publication criteria as it currently stands. Therefore, we invite you to submit a revised version of the manuscript that addresses the points raised during the review process.

**Thank you for your patience with this mamuscript.  I sent it to over 20 potential reviewers, but only one has responded with a critique as well as the statistical referee.  I read through the paper and agree with the assessment of the reviewer with a decision for revision along the lines you will see below.**

We look forward to receiving your revised manuscript.

Kind regards,

Steven E. Wolf, MD

Academic Editor

PLOS ONE

3. Please ensure that you refer to Figure 2, 3, 4, and 5 in your text as, if accepted, production will need this reference to link the reader to the figure.

Additional Editor Comments (if provided):

Reviewers' comments:

Reviewer's Responses to Questions

**Comments to the Author**

1. Is the manuscript technically sound, and do the data support the conclusions?

Reviewer #1: Yes

Reviewer #2: Yes

2. Has the statistical analysis been performed appropriately and rigorously? 

Reviewer #1: Yes

Reviewer #2: Yes

3. Have the authors made all data underlying the findings in their manuscript fully available?

Reviewer #1: Yes

Reviewer #2: Yes

4. Is the manuscript presented in an intelligible fashion and written in standard English?

Reviewer #1: Yes

Reviewer #2: Yes

5. Review Comments to the Author

Reviewer #1: As the statistical reviewer I will focus on methods and reporting. This is generally and well reported study. The main model is appropriate.

Major

1) the power calculation should be presented before the "study participants" section.

2) in the methods section clarify which time points were assessed - was there a primary endpoint? 3 months, 6months? or 9 months. Also how was that factored into the mixed effects model? a separate model for each endpoint, controlled for baseline (arguably the clearest way to analyse and present results from). Alternatively you can have a repeated measures design, which is not what you have (or did not describe as you should have) which may give an "overall" effect across all endpoints.

3) I couldn't replicate the power calculations and i followed the reference the authors provide but i could not tally what was said there to what the authors say (in terms of the SD for example). Nevertheless, the power calculations are correct (on the assumptions provided) but they tend to ignore the potential intraclass correlation coefficient since across many centres. so the authors need to discuss this as a potential problem when it comes to power and also confirm in their response where the information on the assumptions was in the cited paper.

Minor

1) multivariable not multivariate (implies multiple outcomes).

2) methods section "tests of difference" is not a clear enough statement. State the exact tests used for each variable type.

3) the authors say "Model selection was undertaken with each outcome using standard selection heuristics. Covariates were selected based on improving the overall efficiency of the model. Regardless, baseline value of the outcome variable, age, gender, time since injury (1-3 months/4-12 months) and study center were included as covariates in the mixed effects model". The authors need to clarify the model building more clearly in terms of the covariate selection process and explain better how they arrived at the final model.

Reviewer #2: Thank you for the opportunity to review this manuscript. Overall, the study is well-designed and addresses an important clinical question regarding the efficacy and safety of MLC901 (NeuroAiD II) for treating cognitive impairment following mild traumatic brain injury (TBI).

Strengths:

The randomized, double-blind, placebo-controlled, multi-center design is a significant strength and adds considerable value to the research. This approach represents the gold standard for clinical trials, ensuring high internal validity. The use of stratified permuted block randomization and intention-to-treat analysis further enhances the reliability of the results.

The study’s comprehensive evaluation, which includes both primary and secondary outcomes—cognitive functioning, post-concussion symptoms (RPQ), quality of life (QOLIBRI), and the Hospital Anxiety and Depression Scale (HADS)—provides a thorough assessment of the intervention’s effects. Additionally, the detailed reporting of adverse events contributes to the evaluation of the treatment's safety profile.

Achieving a 98% retention rate at the 9-month follow-up is commendable for a study of this length and reinforces the credibility of the data.

Although the primary outcome did not reach statistical significance, the clinically and statistically significant improvements observed in secondary measures—such as post-concussion symptoms, quality of life, anxiety, and depression—are noteworthy and provide valuable insights.

Areas for Improvement:

The lack of significant findings for the primary outcome (complex attention) may raise some concerns. Strengthening the discussion by exploring potential reasons why the intervention did not impact primary cognitive outcomes would be beneficial. It might also help to consider methodological challenges, such as issues with implementing the CNS Vital Signs tool in the Russian context.

Since the CNS Vital Signs tool may not have been fully adapted for use in Russia, this could have influenced the results. It would be helpful to address how cultural and linguistic factors were considered and how they might be mitigated in future research.

The study's population is limited to Russian participants, which may restrict its generalizability. Future studies could benefit from including a more diverse sample to broaden the applicability of the findings.

The data availability statement mentions that some restrictions will apply. For a journal like PLOS ONE, a more open data-sharing policy is typically encouraged. If full data sharing is not possible, a clearer explanation of the restrictions would be beneficial.

The interpretation of the results, particularly the significant improvements in secondary outcomes, could be expanded to provide greater insight into the clinical relevance of MLC901. Even if its cognitive benefits were not demonstrated, the improvement in quality of life and other outcomes is important.

The manuscript could also benefit from some reorganization to enhance clarity, particularly in the results and discussion sections. A more concise presentation of tables and figure legends would help readers better understand the data.

While the discussion covers relevant background information, it would be strengthened by a more thorough comparison with existing literature, particularly other studies on MLC901 and treatments for cognitive impairments following TBI.

Lastly, expanding the discussion of limitations and offering more concrete suggestions for future research would enhance the paper. For example, recommendations for adapting cognitive testing tools to different cultural contexts and exploring the long-term effects of MLC901 on non-cognitive outcomes would be valuable additions.

6. PLOS authors have the option to publish the peer review history of their article (what does this mean? ). If published, this will include your full peer review and any attached files.

**Do you want your identity to be public for this peer review?** For information about this choice, including consent withdrawal, please see our Privacy Policy .

Reviewer #1: No

Reviewer #2: No

---

## [Author Response · Author response to Decision Letter 1]

14 Mar 2025

Response: We have thoroughly reviewed the manuscript and made the necessary corrections to make it fully compliant with the PLOS ONE’s style requirements

2. We note that you have indicated that there are restrictions to data sharing for this study. For studies involving human research participant data or other sensitive data, we encourage authors to share de-identified or anonymized data. However, when data cannot be publicly shared for ethical reasons, we allow authors to make their data sets available upon request.

Response: There are legal restrictions on sharing a de-identified data set of this trial, because the data are owned by a third-party organization (Moleac Pte Ltd) who has imposed such restrictions. However, should the need for data sharing arise, the request may be sent to the Moleac Pte Ltd representative, Dr Sylvain Durrleman on <sylvain.durrleman@moleac.com>

3. Please ensure that you refer to Figure 2, 3, 4, and 5 in your text as, if accepted, production will need this reference to link the reader to the figure.

Response: the text has been revised to ensure that unequivocal reference to each of the tables and figures is made in the manuscript

5. Review Comments to the Author

Reviewer #1: As the statistical reviewer I will focus on methods and reporting. This is generally and well reported study. The main model is appropriate.

Major

1) the power calculation should be presented before the "study participants" section.

Response: the order of the sections has now been revised accordingly. The statistical analyses and power calculation” section has now been split in two sections : “Power calculation” , which is positioned before the “Study participants” section and “Statistical analyses” section, which is the last paragraph of the “METHODS” chapter.

2) in the methods section clarify which time points were assessed - was there a primary endpoint? 3 months, 6 months? or 9 months. Also how was that factored into the mixed effects model? a separate model for each endpoint, controlled for baseline (arguably the clearest way to analyse and present results from). Alternatively, you can have a repeated measures design, which is not what you have (or did not describe as you should have) which may give an "overall" effect across all endpoints.

Response: The primary timepoint was the 6-month assessment for each of the outcomes, corresponding to the end-of-treatment visit. This is clearly stated in the manuscript (line 120). Separate mixed models were constructed for each of the outcome parameters and their dependent variable was the respective outcome parameter change from baseline. The analysis used a repeated measure design with all post baseline timepoints used as repeated effect in each model, while treatment group and its interaction with timepoints along with parameter baseline value, participant age, gender, time since injury (dichotomized into two groups: 1-3 months/4-12 months) and study centers were included as fixed effects in each model. Estimates obtained from the model (Least Squares Means) for the parameters change from baseline at month 6 (primary timepoint) and month 9 were provided with their associated 95%CI and p-values to conclude about treatment efficacy. We added this explanation in the Statistical analysis section of the revised manuscript.

3) I couldn't replicate the power calculations and I followed the reference the authors provide but I could not tally what was said there to what the authors say (in terms of the SD for example). Nevertheless, the power calculations are correct (on the assumptions provided) but they tend to ignore the potential intraclass correlation coefficient since across many centers. So the authors need to discuss this as a potential problem when it comes to power and also confirm in their response where the information on the assumptions was in

the cited paper.

Response: the sample size calculation for the trial was based upon the results obtained in the pilot study done in New Zealand by Theadom et al. (Ref. 16). In this publication, the following data were provided in Supplementary Table 1

Complex Attention score : Mean (SD)

MLC-901 (n=36) Placebo (n=42)

Baseline 83.86 (22.06) 85.43 (16.52)

Month 6 94.18 (15.29) 82.31 (21.56)

(extract from the Supplementary Table 1 in the publication)

As can be seen in the table above, the SD for the score at M6 is in the range 15-21, so the SD for the change from baseline at M6 is likely to be < 20.

However, in view of some uncertainty about the variability of the measure, and in order to be conservative and achieve sufficient power for the trial, we designed the study protocol using a SD of 20.

As pointed out by the Reviewer, the intraclass correlation for centers was not taken into account in the power calculation, as there was limited information to use as to which magnitude of correlation to consider in the calculation. In the final analysis, however, the intraclass correlation for centers is accounted for, via including study centers as fixed effect in the model.

It can be noted also that the withdrawal rate of 30% planned for in the power calculation turned out to be largely overestimated, resulting in a power for the trial higher than the originally planned 80%. We added this information to the Supplementary materials (page 3).

Minor

1) multivariable not multivariate (implies multiple outcomes).

Response: we thank the reviewer for the remark and will make the modifications

2) methods section "tests of difference" is not a clear enough statement. State the exact tests used for each variable type.

Response: we have clarified in the manuscript that t-tests and Chi-2 tests were used for the quantitative and the categorical variables, respectively.

3) the authors say "Model selection was undertaken with each outcome using standard selection heuristics. Covariates were selected based on improving the overall efficiency of the model. Regardless, baseline value of the outcome variable, age, gender, time since injury (1-3 months/4-12 months) and study center were included as covariates in the mixed effects model". The authors need to clarify the model building more clearly in terms of the covariate selection process and explain better how they arrived at the final model.

Response: we thank the reviewer for the remark, and we have simplified the section to make it more precise, as in fact the “heuristic” covariate selection model was more straightforward, and simply included in the model the clinically important variables as covariates. Indeed, as explained in Response to question 2 and stated in the revised manuscript (lines 221-227), the model that was used is a repeated measure design with all post baseline timepoints used as repeated effect, while treatment group and its interaction with timepoints along with parameter baseline value, participant age, gender, time since injury (1-3 months/4-12 months) and study centers were included as fixed effects in each model.

Reviewer #2: Thank you for the opportunity to review this manuscript. Overall, the study is well-designed and addresses an important clinical question regarding the efficacy and safety of MLC901 (NeuroAiD II) for treating cognitive impairment following mild traumatic brain injury (TBI).

Strengths:

The randomized, double-blind, placebo-controlled, multi-center design is a significant strength and adds considerable value to the research. This approach represents the gold standard for clinical trials, ensuring high internal validity. The use of stratified permuted block randomization and intention-to-treat analysis further enhances the reliability of the results.

The study’s comprehensive evaluation, which includes both primary and secondary outcomes—cognitive functioning, post-concussion symptoms (RPQ), quality of life (QOLIBRI), and the Hospital Anxiety and Depression Scale (HADS)—provides a thorough assessment of the intervention’s effects.

Additionally, the detailed reporting of adverse events contributes to the evaluation of the treatment's safety profile.

Achieving a 98% retention rate at the 9-month follow-up is commendable for a study of this length and reinforces the credibility of the data.

Although the primary outcome did not reach statistical significance, the clinically and statistically significant improvements observed in secondary measures—such as post-concussion symptoms, quality of life, anxiety, and depression—are noteworthy and provide valuable insights.

Response: we thank the reviewer for these comments and the recognition of the study’s strengths.

Areas for Improvement:

The lack of significant findings for the primary outcome (complex attention) may raise some concerns. Strengthening the discussion by exploring potential reasons why the intervention did not impact primary cognitive outcomes would be beneficial. It might also help to consider methodological challenges, such as issues with implementing the CNS Vital Signs tool in the Russian context.

Response: we have emphasized in the discussion the methodological challenges and some limitations of the computerized assessment of CNS-VS of Complex Attention and other CNS-VS cognitive parameters.

Since the CNS Vital Signs tool may not have been fully adapted for use in Russia, this could have influenced the results. It would be helpful to address how cultural and linguistic factors were considered and how they might be mitigated in future research.

Response: Indeed, the tool was not fully validated in its Russian translation, and some limitations to travel to Russia, some restrictions due to Covid-19 pandemia leading to training sessions conducted only via virtual meetings, limited investigators meetings, etc. The limited digital literacy of some patients in Russia has also led to suboptimal use of the computerized assessment tool. These methodological challenges and some limitations of the computerized assessment of CNS-VS of Complex Attention and other CNS-VS cognitive parameters in Russia (including cultural and linguistic factors) may have influenced the negative results of these outcome measures in the trial. We added these statements to the Discussion section of the revised manuscript.

The study's population is limited to Russian participants, which may restrict its generalizability. Future studies could benefit from including a more diverse sample to broaden the applicability of the findings.

Response: We add this point as a limitation in the Discussion section, recognizing that, although being a large and multicenter trial, the SAMURAI study is conducted in only one country.

The data availability statement mentions that some restrictions will apply. For a journal like PLOS ONE, a more open data-sharing policy is typically encouraged. If full data sharing is not possible, a clearer explanation of the restrictions would be beneficial.

Response: There are legal restrictions on sharing a de-identified data set of this trial, because the data are owned by a third-party organization (Moleac Pte Ltd) who has imposed such restrictions. However, should the need for data sharing arise, the request may be sent to the Moleac Pte Ltd representative, Dr Sylvain Durrleman on <sylvain.durrleman@moleac.com>

The interpretation of the results, particularly the significant improvements in secondary outcomes, could be expanded to provide greater insight into the clinical relevance of MLC901. Even if its cognitive benefits were not demonstrated, the improvement in quality of life and other outcomes is important.

Response: We have made modifications in the manuscript to further discuss these important endpoints and further emphasize their importance. Specifically, we stated that the significant improvements in non-cognitive secondary outcomes up to 9 months post-randomisation are of considerable clinical significance because up to 20% of people with mild TBI show persistent post-concussion symptoms, about one third of people with mild TBI experience reduced quality of life, and many of them suffer from anxiety or depression.

The manuscript could also benefit from some reorganization to enhance clarity, particularly in the results and discussion sections. A more concise presentation of tables and figure legends would help readers better understand the data.

Response: In the revised manuscript, we re-organized the Methods section, revised the Discussion section, and amended Tables 1-2, and Figures 2-6.

While the discussion covers relevant background information, it would be strengthened by a more thorough comparison with existing literature, particularly other studies on MLC901 and treatments for cognitive impairments following TBI.

Response: Unfortunately, apart from the current trial and pilot trial in New Zealand we referred to, there was no other RCTs evaluating treatment effects of MLC901 on cognitive impairments following TBI.

Lastly, expanding the discussion of limitations and offering more concrete suggestions for future research would enhance the paper. For example, recommendations for adapting cognitive testing tools to different cultural contexts and exploring the long-term effects of MLC901 on non-cognitive outcomes would be valuable additions.

Response: We thank the reviewer for this valuable suggestion. In the revised manuscript, we added to the last paragraph of Discussion that adapting cognitive testing tools to different cultural contexts and exploring the long-term effects of MLC901 on non-cognitive outcomes are also warranted.

Response: Done.

---

## [Decision Letter · Decision Letter 1]

A DOUBLE-BLIND, PLACEBO-CONTROLLED, RANDOMIZED, MULTI-CENTRE, PHASE III STUDY OF MLC901 (NEUROAID IITM) FOR THE TREATMENT OF COGNITIVE IMPAIRMENT AFTER MILD TRAUMATIC BRAIN INJURY

PONE-D-24-36443R1

Dear Dr. Feigin,

We’re pleased to inform you that your manuscript has been judged scientifically suitable for publication and will be formally accepted for publication once it meets all outstanding technical requirements.

Kind regards,

Steven E. Wolf, MD

Academic Editor

PLOS ONE

Additional Editor Comments (optional):

Reviewers' comments:

Reviewer's Responses to Questions

**Comments to the Author**

1. If the authors have adequately addressed your comments raised in a previous round of review and you feel that this manuscript is now acceptable for publication, you may indicate that here to bypass the “Comments to the Author” section, enter your conflict of interest statement in the “Confidential to Editor” section, and submit your "Accept" recommendation.

Reviewer #1: All comments have been addressed

Reviewer #2: All comments have been addressed

2. Is the manuscript technically sound, and do the data support the conclusions?

Reviewer #1: Yes

Reviewer #2: Yes

3. Has the statistical analysis been performed appropriately and rigorously? 

Reviewer #1: Yes

Reviewer #2: Yes

4. Have the authors made all data underlying the findings in their manuscript fully available?

Reviewer #1: Yes

Reviewer #2: Yes

5. Is the manuscript presented in an intelligible fashion and written in standard English?

Reviewer #1: (No Response)

Reviewer #2: Yes

6. Review Comments to the Author

Reviewer #1: I am satisfied with the authors' responses and the resulting changes to the manuscript. Nothing else to add.

Reviewer #2: The authors have satisfactorily addressed all of my prior concerns and clarified all points raised in my review.

7. PLOS authors have the option to publish the peer review history of their article (what does this mean? ). If published, this will include your full peer review and any attached files.

**Do you want your identity to be public for this peer review?** For information about this choice, including consent withdrawal, please see our Privacy Policy .

Reviewer #1: No

Reviewer #2: No

---

## [Editor Report · Acceptance letter]

PONE-D-24-36443R1

PLOS ONE

Dear Dr. Feigin,

I'm pleased to inform you that your manuscript has been deemed suitable for publication in PLOS ONE. Congratulations! Your manuscript is now being handed over to our production team.

Kind regards,

on behalf of

Dr. Steven E. Wolf

Academic Editor

PLOS ONE